# Contraceptive-induced menstrual changes in low- and middle-income countries: a systematic scoping review

Maureen Makama [1,2] ✉, Annie R. A. McDougall[1,3], Jenny Jung[1], Fiona Bruinsma[1], Elly Layton[1], Jenny Cao[1], Kate Mills[1], Tahlia R. Guneratne[1], Paige Riddington[4,5], Anne Ammerdorffer [6], Luis Bahamondes[7], Lorena Romero [8], Jen Sothornwit [9], Pisake Lumbiganon[9], A. Metin Gülmezoglu[6] & Joshua P. Vogel[1,3]

## Abstract

**Background** Contraceptive-induced menstrual changes (CIMCs) contribute substantially to women's dissatisfaction with and discontinuation of contraceptives. We summarised evidence on the prevalence, health impact, treatment, and barriers to accessing treatment for CIMC in low- and middle-income countries (LMICs).
**Methods** Nine databases (MEDLINE, Embase, Emcare, PsycINFO, Global Health, Global Index Medicus, CINAHL, Web of Science, and Scopus) were systematically searched for studies published from January 1, 2000, to December 16, 2024. Eligible studies included reproductive-age women (15–49 years) using any modern contraceptive (excluding barrier and permanent methods) in LMICs. Findings were categorised according to the World Health Organization's Belsey definitions of frequency and severity of CIMC-related bleeding. Quantitative data were summarised using descriptive statistics and qualitative data using thematic synthesis.
**Results** Here we include 321 studies conducted in 44 countries. The prevalence of CIMCs range from 0–94% and vary by contraceptive type. Two-fifths (40.2%) of the prevalence reports did not define the type of CIMC experienced by participants. The most frequently reported health impact of CIMCs is contraceptive discontinuation leading to an unmet need for contraception. Non-steroidal anti-inflammatory drugs are the most frequently investigated treatment. No studies report on the barriers to accessing treatment for CIMCs in LMICs.
**Conclusion** CIMC impacts contraceptive users in various ways depending on the contraceptive type and user's perception of it, highlighting the importance of counselling. Primary research must use standardised definitions of CIMC to improve data quality. Investment in research and development of innovative therapeutics and novel approaches to reducing CIMC is needed to mitigate the unmet need for contraception in LMICs.

## Plain language summary

Contraceptives are essential for preventing unplanned pregnancies. However, users may experience changes in their menstrual cycle which may cause dissatisfaction. This study mapped out the most recent research on menstrual changes caused by contraceptives–including how often it occurs, how it impacts users, how it is treated and what hinders treatment–in low and middle-income countries. Our results show that the frequency of menstrual changes caused by contraceptives differed by contraceptive type. These changes cause many women to stop contraceptives making them more likely to have unplanned pregnancies. This high-lights the importance of counselling contra-ceptive users and the need to develop new contraceptive methods that have less influence on the menstrual cycle.

Contraceptive prevalence using any modern method in women (15–49 years) varies widely between countries, with the lowest rates in low-income countries (31%), followed by lower-middle-income countries (47%), and upper-middle-income countries (72%)[1–4]. There are significant health and financial consequences associated with unintended pregnancies, including higher rates of unsafe abortions, increased maternal and child morbidity and mortality, reduced workforce participation, and lower household incomes[5–7]. Adolescent pregnancies can be particularly harmful, leading to girls dropping out of school, lower educational attainment and poverty, in addition to high rates of pregnancy and birth complications[8].

For most women, a normal menstrual cycle occurs every 21 to 35 days with bleeding lasting from 2–7 days[9]. Menstrual changes may include

changes in the frequency, regularity of onset, duration of flow, and volume of blood from the normal menstrual cycle as classified by the International Federation of Gynecology and Obstetrics (FIGO) system for nomenclature of normal and abnormal uterine bleeding[10,11]. Contraceptive-induced menstrual change (CIMC) is defined as any menstrual change experienced as a direct adverse effect of contraceptive use[12]. Many contraceptives influence the menstrual cycle and can allow for continued cyclic bleeding (i.e., normal bleeding patterns), or result in partial or complete suppression of the normal cycle[13–16]. Using a combined hormonal contraceptive allows for continued cyclic bleeding, though users may experience breakthrough bleeding (spotting or bleeding between periods), while long-acting reversible contraceptives (LARC) can cause menstrual bleeding to cease completely[14].

Adherence to and continuation of modern contraceptive use is necessary to ensure their effectiveness in preventing pregnancy[5]. However, adverse effects of contraceptives are a major cause of discontinuation or poor compliance with contraceptive use, which increases contraceptive failure rates[17]. CIMCs have been shown to impact compliance and women's willingness to use contraceptives[17–19]. A 2019 Cochrane review reported that 32% of women on short-acting hormonal contraceptives are likely to discontinue their use due to menstrual disturbances[5]. A retrospective analysis of Demographic and Health Surveys (DHS) data from 36 low- and middle-income countries (LMICs) from 2005 to 2014 reported that 41% of women who last used a short-acting hormonal contraceptive and 40% who last used a LARC method discontinued use due to side effects and health concerns[20].

Despite the suboptimal prevalence of modern contraceptive use in many LMICs, the prevalence of CIMC and its impact on contraceptive discontinuation in these settings have not been systematically synthesised. Also, the accessibility of treatment and management strategies for CIMC in LMICs is unknown. This review aimed to summarise the evidence on the prevalence and health impact of CIMC across LMICs; the current treatments used for its management globally; and the barriers to accessing these treatments in LMICs.

In this review, we find that the prevalence of CIMC ranges widely, varying by contraceptive type. A significant proportion of the prevalence reports do not specify the type of CIMC experienced. Contraceptive discontinuation leading to unmet need for contraception is the most frequently reported health impact of CIMC. Non-steroidal anti-inflammatory drugs are the most frequently investigated treatment for CIMC, and no studies report on barriers to accessing treatment for CIMCs in LMICs.

## Methods

We conducted the scoping review following the six-stage process developed by Arksey and O'Malley and further described by Levac et al.[21–23]. The findings of this review are reported using the Preferred Reporting Items for Systematic Reviews and Meta-Analyses Extension for Scoping Reviews (PRISMA-ScR) checklist[24] (Supplementary Data 1). We registered the protocol online on Open Science Framework, (registration identifier **u5jr8**) available at https://doi.org/10.17605/OSF.IO/U5JR8.

### Statistics and reproducibility

**Search strategy and selection criteria.** The search strategy was formulated using the Participants/Population, Concept and Context (PCC) framework[25]. The participants/population were reproductive-age women (15–49 years) using any form of modern contraceptive (excluding barrier and permanent methods) as defined by the source articles. The concepts explored included the prevalence, health impact, treatment and barriers to accessing treatment. The context considered included studies in countries classified as low-income, lower-middle-income or upper-middle-income according to the 2022 World Bank classification of countries by income level[26]. For studies providing data on treatment options for CIMC, we included studies regardless of country. We included primary research using the following study designs: interventional trials (randomised and non-randomised trials); observational studies (prospective cohort, retrospective, cross-sectional, and case-control studies); qualitative studies; mixed methods studies; and review of primary research (only systematic reviews and meta-analysis). We did not limit by language of publication but used translation services for papers in a language other than English.

We excluded studies relating to menstrual changes not associated with contraceptive use; those exploring emergency contraceptives, barrier, or permanent methods only; or exploring menstrual suppression. Studies using the following study designs were excluded: case studies, case reports, case series; journal articles that do not present primary data (such as commentaries, letters to the editor, opinion articles); dissertations, conference abstracts, clinical guidelines, clinical trial protocols/registries; other review types (narrative, rapid, and scoping reviews). We also excluded studies where full text articles could not be retrieved.

We developed a search strategy in consultation with an information specialist (LR). We combined search terms relating to contraceptives and menstrual changes and LMICs using search terms related to prevalence, health impact, and barriers to accessing treatment (Supplementary Table S1). When searching for studies related to CIMC treatments, we excluded LMICs from the search terms. We searched nine databases (MEDLINE, Embase, Emcare, PsycINFO, and Global Health (via Ovid); Global Index Medicus; CINAHL via EBSCOhost; Web of Science; and Scopus) for studies published between January 1, 2000 and December 16, 2024 (date of search). We limited the search to studies published from 2000 onwards to ensure the review captured the most recent and relevant evidence. This timeframe reflects a period of significant innovation in contraceptive technologies, including the wider adoption of long-acting reversible contraceptives. While this approach may exclude some earlier seminal studies, our focus was on synthesising evidence aligned with current clinical practice and policy priorities[27,28].

**Study selection.** Two reviewers (MM, JJ, FB, EL, AM, KM, PR or AA) independently screened each title and abstract. After retrieving full texts of potentially eligible studies, two reviewers (MM, LB, JJ, EL, KM, PR, FB, AM, JC) independently screened each article. Any discrepancies in eligibility assessment were resolved by discussion within the review team. All stages of the screening were managed using Covidence software[29].

**Data extraction and analysis.** According to the World Health Organization (WHO) Belsey criteria, changes in bleeding patterns with contraceptive use can be categorised as: amenorrhoea (no bleeding or spotting during a 90-day reference period), prolonged bleeding (bleeding or spotting episodes lasting more than 14 days during a 90-day reference period), frequent bleeding (more than 5 bleeding or spotting episodes during a 90-day reference period), infrequent bleeding (1 or 2 bleeding or spotting episodes during a 90-day reference period), and irregular bleeding (3–5 bleeding episodes and less than 3 bleeding or spotting-free intervals of 14 days or more during a 90-day reference period)[30,31]. These criteria were used for data analysis; however, when definitions were not specified data were categorised as defined by study authors.

A standardised data extraction template was developed in Covidence. The data extracted included characteristics of each study (study designs, country, year); type of contraceptive used, and the data reported (prevalence, health impact, treatment and/or barriers to accessing treatment of CIMCs). Data extraction was conducted independently by any two reviewers (MM, JJ, EL, KM, PR, TG, FB, JC). First, double data extraction was conducted for 82 (25.5%) studies and consensus reached; then single data extraction and cross-checking were done independently for the remaining studies. We employed an explanatory sequential synthesis method to integrate quantitative and qualitative findings. First, quantitative findings were synthesised using descriptive statistics (frequency and percentages) in Microsoft Excel to summarise key trends and patterns. Next, following previously published methods, two reviewers (MM and JC) conducted a thematic synthesis of qualitative findings. This involved line-by-line coding of the extracted data, which were organised into initial codes

and iteratively refined to inductively generate and define overarching themes[32]. The qualitative findings were then used to elaborate on and contextualise the quantitative findings. Integration was conducted through a narrative approach, highlighting areas where qualitative insights clarified or explained trends and patterns observed in the quantitative data. To avoid duplication of data, systematic reviews were excluded from the main analysis. However, we reported the number of systematic reviews addressing each of the key concepts to provide an overview of the evidence base.

## Results

A total of 17,672 records from nine databases were retrieved (Fig. 1). After duplicates were removed 10,037 records were screened at title/abstract stage and 493 were included for full-text screening. Overall, 16 papers could not be retrieved, and 156 articles were excluded at full-text screening (Supplementary Data 2), leaving 321 unique studies for inclusion (Supplementary Data 3).

### Characteristics of included studies

There were 42 (13.1%) studies published from 2000 to 2004; 61 (19.0%) from 2005 to 2009; 59 (18.4%) from 2010 to 2014; 57 (17.8%) from 2015 to 2019; and 102 (31.8%) from 2020 to the date of the search. Of the 321 included studies, two-thirds were observational studies (214 studies). This included prospective cohorts (119 studies, 37.1%), retrospective studies (48 studies, 15.0%), cross-sectional studies (44 studies, 13.7%), and a case-control study (3 study, 0.9%). Sixty-two interventional studies included randomised controlled trials (52 studies, 16.2%) and non-randomised trials (9 studies, 2.8%). The remaining studies were qualitative (19 studies, 5.9%),

mixed-methods (11 studies, 3.4%), and systematic reviews (16 studies, 5.0%). Prevalence and health impact data were found across all types of study designs. Treatment data were only reported in systematic reviews, randomised trials, non-randomised trials, and prospective cohort studies (Supplementary Fig. S1). No studies were identified that reported on the barriers to treatment of contraception-induced menstrual irregularities. Of the 16 systematic reviews included, nine (56%) reported on treatment, four (25%) reported on health impact and three (19%) reported on the prevalence of CIMC.

The studies reporting data related to the prevalence and health impact of CIMC took place in 44 countries, of which 14 countries had only one study (Supplementary Fig. S2a, Supplementary Table S2). The largest number of studies took place in India (43 studies) and Nigeria (37 studies). In total, 56% (166/297) of prevalence and health impact studies were conducted in lower-middle-income countries, 37% (111/297) in upper-middle-income countries and 7% (20/297) in low-income countries (Supplementary Fig. S2b). For data related to CIMC treatment, 44% (16/36) of studies were conducted in lower-middle-income countries, 31% (11/36) in upper-middle-income countries and 25% (9/36) in high-income countries (Supplementary Fig. S3). There were no studies in low-income countries that reported on CIMC treatment.

### Types of contraceptives in the included studies

The types of contraceptives used in the included studies are described in Supplementary Table S3. Implants and intrauterine devices were the most often reported contraceptives, each making up 28.2% (60 studies). Other types included progestin-only contraceptives (47 studies, 22.1%); combined

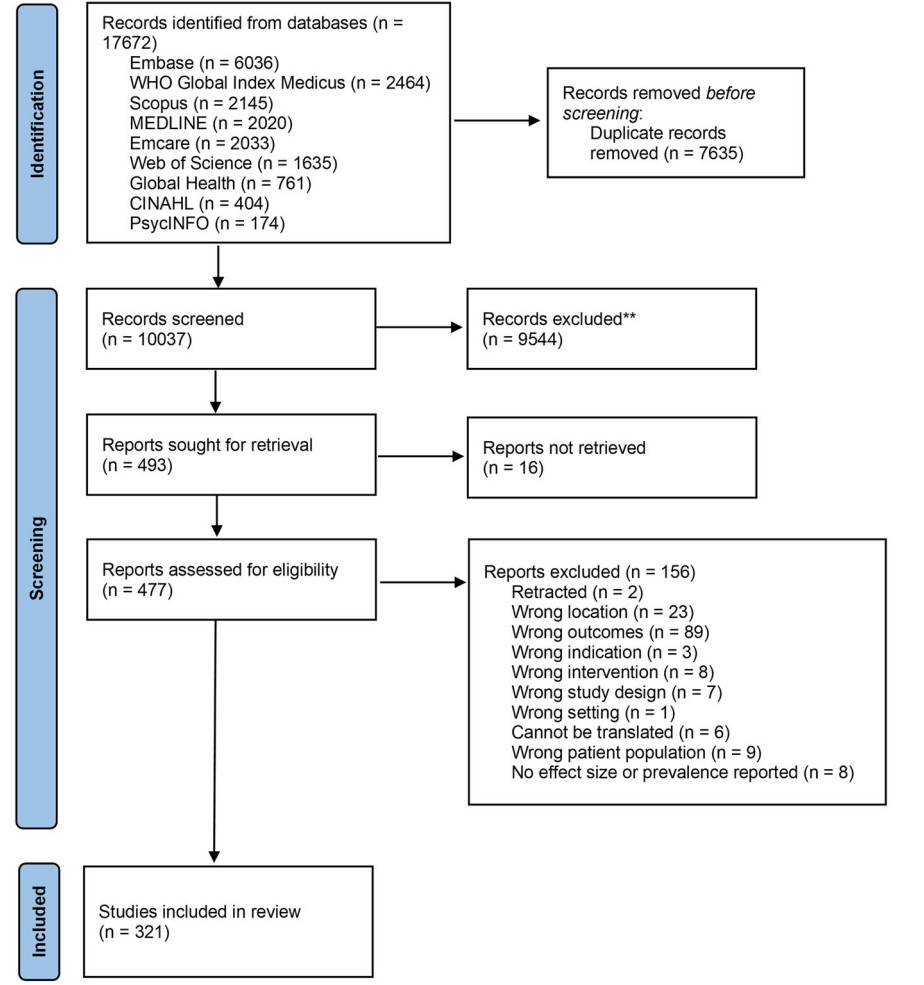

**Fig. 1 | PRISMA flow diagram of systematic literature search.** Flow diagram illustrating the identification, screening, eligibility assessment, and inclusion of studies in the scoping review.

hormonal contraceptive (29 studies, 13.6%); selective oestrogen receptor modulator (Ormeloxifene) (3 studies, 1.4%); and other unspecified contraceptives (14 studies, 6.6%).

### Prevalence of contraceptive-induced menstrual changes

Of the 321 included studies, 220 primary studies reported data on the prevalence of CIMC, constituting 521 reports of menstrual changes (Table 1). The number of participants in these studies ranged from 20 to 9262. The overall prevalence of CIMC ranged widely from 0 to 94%, though many reports (233/580, 40.2%) did not define the type of menstrual change experienced. Infrequent bleeding was the least prevalent type of CIMC reported (0 to 60.9%). Studies with a follow-up period of at least one year (106 studies, 47.7%) reported lower prevalence ranges across all types of CIMC. Twenty studies (9.0% of all studies reporting prevalence) included at least 1000 participants and prevalence ranges were narrower. Supplementary Table S4 shows the prevalence by type of contraceptive used and type of CIMC reported. Prevalence varied by contraceptive type; CIMC was more common in users of intrauterine devices and least common in users of combined hormonal contraceptives.

### Health impact of contraceptive-induced menstrual changes

We identified 125 primary studies that reported the health impact of CIMC. The number of participants included in these studies ranged from 10 to 9262 participants. Contraceptive discontinuation, reported by 98 (78.4%) studies, was the most frequently reported health impact. However, the proportion of contraceptive discontinuation attributable to CIMC varied across contraceptive types (Supplementary Table S5). Users of intrauterine devices had the highest proportion of discontinuation due to CIMC (up to 100%) and users of ormeloxifene had the lowest proportion of discontinuation (30% from one study) followed by combined hormonal contraceptives (up to 71.9%). The health impacts of CIMC reported by 27 qualitative and mixed-methods studies were synthesised into six themes described below and in Table 2.

### Increased economic burden.

This included increased economic burden on families due to healthcare costs incurred by unanticipated visits to the hospital and increased need to purchase sanitary products[33,34]. Women also reported that CIMC impeded their ability to work and earn a living due to difficulty in moving around when bleeding is heavy[34,35].

### Interference with regular activities.

Women reported difficulty in completing everyday household tasks or attend work or school due to fatigue and discomfort from heavy menstrual bleeding[36]. Women reported need for constant washing, which is particularly challenging for those with limited access to water[37]. Prolonged and irregular bleeding patterns prevented engagement in religious practices like prayers or handling religious books[38–41].

*Experiences of social stigma, shame and exclusion*: Women reported experiencing stigma and exclusion due to sociocultural norms and religious beliefs that perceive menstruation as unclean[38,42]. Some women reported experiencing feelings of shame when their family notice the excessive laundry or when they accidentally bleed in the place of prayer[42]. Women reported not feeling good about themselves when they were bleeding heavily or for prolonged periods of time[37].

### Unmet need for contraception.

Many women discontinued a contraceptive method due to menstrual irregularity[43–45]. Heavy and prolonged menstrual flow was the most often reported reason for discontinuing contraceptives, with fewer women discontinuing because of amenorrhoea[33,35,42,46]. When women discontinued a type of contraceptive, they were often reluctant to start or delay starting another contraceptive method, thus increasing the likelihood of unplanned pregnancy[35,47]. This was also the most frequently reported impact of CIMC from quantitative studies.

### Psychological impact.

Women reported experiencing psychological stress due to fear of the menstrual side effects of contraceptives, including fear of being pregnant when amenorrhea and fear of expulsion of intrauterine device due to heavy bleeding[48]. They reported being self-conscious and worried about the possibility of staining their clothes, which was considered shameful[42]. There were women who had feelings of guilt because they could not contribute to their household tasks such as cooking or tending to livestock because of heavy bleeding[42].

### Contributes to marital problems.

Women reported having problems with their partners due to sexual abstinence because of persistent bleeding[34,38]. This resulted in sexual dissatisfaction and increased verbal and physical abuse by their partners[34,38,42]. Some were concerned that because prolonged bleeding prevented them from having sex, their partners will seek sexual satisfaction elsewhere, making them vulnerable to sexually transmitted disease[38].

*Health concerns:* Six studies reported health concerns due to CIMCs[35,39,40,49–51]. There were concerns that heavy bleeding could lead to health issues such as anaemia and exacerbate pre-existing medical conditions[35]. In particular, women with limited access to quality and nutritious foods worried that the combination of poor nutrition and increased menstrual bleeding would be detrimental to their health[49].

### Treatment of contraceptive-induced menstrual changes

We identified 36 primary studies reporting on the treatment of CIMC. There were 22 individual drugs reported in the studies identified, 14 of which were only reported by one study (Supplementary Table S6). The most frequently investigated drugs for the treatment of CIMC were non-steroidal anti-inflammatory drugs (18 studies) followed by the antifibrinolytic drug tranexamic acid (9 studies). Other drugs reported were antiprogesterones (5 studies), antibiotics (3 studies), oestrogen (4 studies), combined progestin and oestrogen (3 studies), dietary supplements (3 studies), vitamins

**Table 1 | Prevalence of contraceptive-induced menstrual changes reported (222 studies)**

| Type of menstrual change | Number of reports[a] | Sample size | Prevalence reported | Prevalence in studies with ≥ 1 year follow-up (106 studies) | Prevalence in studies with ≥ 1000 participants (20 studies) |
|---|---|---|---|---|---|
| Heavy menstrual bleeding | 74 | 21–9262 | 0–64.3% | 0–52.7% | 1.8–44.1% |
| Breakthrough (intermenstrual) bleeding/spotting | 53 | 23–9262 | 0–70.0% | 0–50.0% | 3.4–27.9% |
| Amenorrhoea | 110 | 23–9262 | 0–80.0% | 0–80.0% | 3.5–80.0% |
| Frequent bleeding | 25 | 23–1994 | 0–63.6% | 0–17.4% | 5.1–55.2% |
| Infrequent bleeding | 30 | 23–9262 | 0–60.9% | 0–39.1% | 0–24.2% |
| Prolonged bleeding | 40 | 20–1868 | 0–78.9% | 0–58.0% | 0–44.8% |
| Menstrual changes (unspecified) | 231 | 20–9262 | 0–94.0% | 0–81.3% | 0–80.0% |

[a]Does not add up to the total 222 studies because studies reported more than one type of menstrual change.

**Table 2 | Thematic analysis of the health impact of contraceptive-induced menstrual changes**

| Themes | Description | Illustrative quotes | References |
|---|---|---|---|
| Increased economic burden | Bleeding impeded their ability to work and earn a living; CIMC resulted in extra healthcare costs due to unanticipated visits to the hospital; increased need for sanitary pads | *'I haven't been able to work for the past three weeks as the bleeding is heavy and moving about is difficult.'* | 34,35,37,68,69 |
| Interferes with regular activities | Excessive bleeding interfered with daily life activities by causing fatigue and an inability to complete household tasks; women experienced discomfort due to bleeding; women had to wash always to feel clean; missed school because of heavy periods | *'Because of physical weakness and dizziness, I could not manage all the domestic work properly.'* | 35–40,42,68,70 |
| Experiences of social stigma, shame and exclusion | Prolonged and/or irregular bleeding interrupted religious practices; religious and cultural beliefs prevented prayers, sexual intercourse and attending gatherings during bleeding | *'… but when I started using it, I didn't know how to pray anymore. Because I don't know when I'm going to finish my period and I can resume my prayers.'* | 35,37–39,42 |
| Unmet need for contraception | Menstrual irregularities were a major cause of contraceptive discontinuation | *'I stopped using it because I was menstruating every two weeks.'* | 5,34–36,42,44,45,48,51,54,59,61,71–75 |
| Psychological impact | Concern about the possibility of infertility; fear of side effects; worry about pregnancy when amenorrhoea was experienced; fear of expulsion of IUD due to excessive bleeding; self-consciousness and feelings of shame due to the possibility of staining clothes; | *'I was mentally upset I could not go to any gathering because I always had to be conscious of my clothes because they might be soaked with blood.'* | 33,42,48,51,75,76 |
| Contributes to marital problems | Cohabitation problems including verbal and physical abuse from partners; sexual abstinence because of bleeding | *'It's the heavy menstruation that makes a man leave his wife'* *'…am removing the implant because my partner and I are no longer having sex like before because of the bleeding, which is not stopping.'* | 38,50,57,69,73,77 |
| Health concerns | Possibility of anaemia due to excessive bleeding | *'I had to remove the implant because my periods became very heavy and I was losing a lot of blood.'* | 35,39,40,49–51 |

(2 studies); and one study each reported the use of an antidiuretic hormone, a selective oestrogen receptor modulator, and a selective progesterone receptor modulator.

### Ethics approval and consent to participate

Ethics approval was not required as no primary data collection was undertaken.

## Discussion

This scoping review is the first to map the evidence on the prevalence, health impact, treatment, and barriers to accessing treatment of CIMC across LMICs. We identified 321 studies that reported on one or more of these concepts. The prevalence of the different types of menstrual changes ranged widely across contraceptive types. The most commonly reported health impact or consequence of CIMC was contraceptive discontinuation, which can worsen the unmet need for contraception. There were a limited number of studies that investigated drugs used for the treatment of CIMC. Non-steroidal anti-inflammatory drugs were the most frequently reported drugs used for the treatment of CIMC. No studies were identified that reported the barriers to accessing treatment for CIMC. The findings build on the existing literature and expand our understanding of the prevalence, health impact, and treatment of CIMC in LMICs. Significant research gaps identified include many primary studies not using standardised definitions for CIMC and short follow-up periods.

We identified many studies that reported the prevalence of CIMC; however, these studies varied widely in the number of participants included (20–9262), the duration of follow-up (1 month to 10 years), and the types of contraceptives used by the participants. Sub-analyses of studies using standard definitions and measurements of CIMC with at least 12 months of follow-up in LMICs yielded less variable prevalence estimates. This highlights the importance of using standardised definitions and measurements of CIMC in primary studies to improve comparison between studies and optimise CIMC data[31]. Optimising data will

support counselling and guide decision making regarding contraceptive choice[31].

Consistent with our findings, a 2018 scoping review of data from all countries, which included only 100 English-language studies, reported substantial variation in how women respond to CIMC[52]. They found that the health impact of CIMC may be influenced by individual and social factors[52,53]. For example, existing social stigma, socio-cultural norms, and religious beliefs surrounding menstruation may influence how women perceive CIMC[12,54]. Our review identified a range of economic, psychological, social, and relational impacts on individuals in LMICs. This was particularly the case with heavier, longer or irregular bleeding which increases discomfort, washing, and consciousness of staining clothes; and inhibits participation in regular activities such as work, religious practices, and sex[35,38,39,42]. CIMC reduces the quality of life of individuals and exacerbates menstrual hygiene management issues (period poverty), including access to water, sanitation, and hygiene facilities, particularly in low-resource settings[12,55]. Contraceptive discontinuation due to CIMC was the most frequently reported health impact of CIMC. Contraceptive discontinuation significantly contributes to an unmet need for contraception to address unintended pregnancies and unsafe abortions, which are more prevalent in lower-resource settings[56–58].

The health impact of CIMC varies depending on personal beliefs, attitudes towards menstruation, and cultural context[59]. For example, amenorrhoea associated with contraceptive use may be viewed as a benefit by some and a problem for others[33,48,60]. Thus, appropriate counselling of clients by healthcare workers could reduce contraceptive dissatisfaction and discontinuation, and decrease the unmet need for contraception[41,56,61]. Previous studies have suggested a link between the quality of care received at contraceptive initiation and contraceptive continuation rates[41,62]. Healthcare providers could better support contraceptive users by obtaining more information regarding their client's perceptions of menstruation and contraception, warning them about side effects and reassuring them about health concerns[63,64].

Although we identified 22 drugs for the treatment of CIMC, the number of studies that investigated these drugs was limited, highlighting a significant research gap. Some of the drugs we identified were also included in a 2013 Cochrane review on medical treatments of bleeding irregularities associated with the use of progestin-only contraceptives[65]. The review concluded that, although promising, there was insufficient evidence to recommend their routine use[65]. Currently, none of the researched treatment options for CIMC are recommended for long-term clinical use[65]. Some have suggested that method switching could be employed for the management of CIMC[66]. Nevertheless, the need to use one drug to treat the side effects of another drug can cause users to abandon contraceptive use altogether. This poses a concern and limits the contraceptive options of many women of reproductive age who may be dissatisfied with the menstrual changes experienced. Research and development into new and innovative contraceptives with negligible effects on the menstrual cycle and/or new therapeutics that can be used for the long-term management of CIMC is therefore paramount[12].

We did not identify any studies that reported barriers to accessing treatment in LMICs. This may be because these drugs are commonly used for the management of other conditions and are already widely available and accessible. It is also possible that there are studies that report barriers to accessing these treatment options outside the context of CIMC. More research is needed to elucidate the most effective interventions and strategies to address CIMC. It is also important to explore users' satisfaction with using these medications to mitigate the side effects of contraceptive use.

Almost half of the included studies did not define the type of menstrual change experienced, which is vital when describing the impact of different contraceptives on the menstrual cycle. For example, for combined hormonal contraceptives, the oestrogen content helps to stabilise bleeding patterns, although breakthrough bleeding occurs in up to 30% of users for the first three to four cycles[66,67]. Breakthrough bleeding is, however, still reported in up to 10% of users after 12 months of use[66]. Therefore, defining the type of menstrual change experienced by participants and having longer follow-ups may reduce variations and provide more useful prevalence data. Furthermore, as the data on CIMC relies mainly on self-reporting, the context in which the questions are asked and individual perceptions of CIMC may influence findings. There is, therefore a need for uniformity in how CIMC is measured. Using standardised and validated measures can improve the reliability and quality of data; therefore, future primary research needs to ensure that standard definitions of CIMC are used[30,31].

Our review has several strengths. First, we used the Arksey and O'Malley Framework to guide the conduct of the review and the PRISMA ScR checklist to report the findings[21,22]. Second, two reviewers independently performed data screening and extraction ensuring the integrity of the process. Third, we did not have language limitations ensuring that studies from non-English speaking countries were captured. Lastly, our review focused on LMICs, which have the largest health impact of unmet need for contraception[56–58].

Some limitations should be considered when interpreting the findings of this review. We did not critically appraise the methodological quality of the included studies. However, this is not a requirement for scoping reviews as our aim was to map and describe the breadth of evidence on CIMC in low and middle-income countries. Furthermore, as our review focuses on descriptive synthesis rather than clinical outcomes, study quality is unlikely to significantly impact on conclusions.

## Conclusion
Our review found that CIMC is a major contributor to contraceptive dissatisfaction and discontinuation. However, CIMC has not been the focus of research amidst efforts to improve the uptake of modern contraception[12]. There is need for research focused on strategies to reduce the health impact of CIMC in low- and middle-income countries. This may include new approaches to contraceptive counselling and new and innovative contraceptives that might reduce CIMC and improve contraceptive satisfaction among clients.

## Data availability
All data generated or analysed in this review are included in this published article and its supplementary information files.

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

## Acknowledgements

This review was funded by the Bill and Melinda Gates Foundation (INV-038938). The views expressed are those of the authors and not necessarily those of the Bill and Melinda Gates Foundation. The funders had no role in the study design, data collection, analysis, interpretation, or writing of the report.

## Author contributions

M.M., A.R.A.M., A.A., L.B., J.S., P.L., A.M.G. and J.P.V. contributed to the conception and design of the review. A.R.A.M., A.A., A.M.G. and J.P.V. contributed to the acquisition of funds. M.M., A.R.A.M. and L.R. developed the search strategy. M.M., A.R.A.M., J.J., F.B., E.L., J.C., K.M., T.R.G., P.R., A.A., and L.B. performed the study selection and data extraction. M.M. and J.C. performed the data analysis. M.M. wrote the first draft of the manuscript. All authors contributed to the interpretation of the data, revision of drafts and approved the final manuscript.

## Competing interests
The authors declare no competing interests.

## Additional information

[1]Women's, Children's and Adolescents' Health Program, Burnet Institute, Melbourne, VIC, Australia. [2]Health and Social Care Unit, School of Public Health and Preventive Medicine, Monash University, Melbourne, VIC, Australia. [3]Monash Institute of Pharmaceutical Sciences, Monash University, Parkville, VIC, Australia. [4]The Ritchie Centre, Hudson Institute of Medical Research, Clayton, VIC, Australia. [5]Department of Obstetrics and Gynaecology, Monash University, Clayton, VIC, Australia. [6]Concept Foundation, Geneva, Switzerland/, Bangkok, Thailand. [7]Department of Obstetrics and Gynaecology, University of Campinas Faculty of Medical Sciences Campinas, Campinas, SP, Brazil. [8]The Ian Potter Library, Alfred Health, Melbourne, VIC, Australia. [9]Department of Obstetrics and Gynecology, Faculty of Medicine, Khon Kaen University, Khon Kaen, Thailand. ✉e-mail: maureen.makama@burnet.edu.au

