## [Transparent Peer Review file · Communications Medicine]

Contraceptive-induced menstrual changes in low- and middle-income countries: a systematic scoping review

Corresponding Author: Dr Maureen Makama

Version 0:

Reviewer comments:

Reviewer #1

(Remarks to the Author)

What are the major claims of the paper? Are they novel and will they be of interest to others in the community and the wider field?

This is a scoping review of the existing literature and as such, none of the claims are 'novel'; however, the review does bring together the evidence to allow an understanding/overview of the current status of the effect of CMIC on contraceptive use in LMICs.

Is the work convincing, and if not, what further evidence would be required to strengthen the conclusions?

It is disappointing that the qualitative data is not presented, apart from a list of themes. As a mixed methods review, the reporting should pay as much attention to the qualitative findings as the quantitative findings and there should also be a further synthesis of how the two paradigms resonate together.

Do you feel that the paper will influence thinking in the field?

I am unsure that the paper will influence thinking in the field, although it does provide an overview of the current situation on which to base future research.

Please feel free to raise any further questions and concerns about the paper.

Abstract: Please clarify your synthesis method. In line 11 delete 'women' as this is duplicated.

Methods:

1. Add the reproductive age range in line 11
2. How did you ensure that there was no double reporting due to the inclusion of systematic reviews in your dataset? Surely some of these 16 reviews would have also included empirical studies you retrieved in your search?
3. Did you use a framework for your search strategy eg PICO?
4. What is the justification for your search start date of 1st January 2000?

Findings:

1. 321 studies is very high and perhaps resulted in the lack of presentation of qualitative findings?
2. The qualitative synthesis needs attention - a list of themes is not appropriate in a published review.

Discussion:

1. The 2009 review discussed in lines 205-211 - is this one of the included reviews in your scoping review or is it meant to be supporting literature? It isn't clear and doesn't seem to resonate with the previous point.
2. Please provide the limitations of your review as well as the strengths
3. Please provide a summary of the implications of your review for clinical practice
4. Please provide a summary of recommendations for future research

Tables:

There is no table of included study characteristics which is good practice in a review

Appropriateness and validity of any statistical analysis?

Descriptive analysis (frequencies and percentages) used.

Ability of a researcher to reproduce the work, given the level of detail provided?

I do not feel that there is sufficient methods detail to reproduce this study.

Reviewer #4

(Remarks to the Author)

The paper reports on findings from a systematic scoping review on menstrual changes associated with contraception. The review draws from a large data set of 321 studies from 44 low- and medium-income countries (LMIC) on contraceptive dissatisfaction and subsequent method discontinuation, and how this contributes to a high unmet need for contraception in LMIC. The prevalence of contraceptive-induced menstrual changes varied from 0-94%, depending by contraceptive type. In this context, the review paper calls for greater investment in research and development of therapeutics and novel approaches to reduce contraceptive-induced menstrual changes.

Changes documented include prolonged or frequent or irregular bleeding,

The findings from the review make a compelling case for addressing contraceptive-induced menstrual changes. From their deduction from the findings, the authors allude to method discontinuation due to contraceptive-induced menstrual changes as the most frequently reported "the burden of contraceptive-induced menstrual changes". Should this rather be "outcome associated with contraceptive-induced menstrual changes?"

Version 1:

Reviewer comments:

Reviewer #1

(Remarks to the Author)

Thank you for submitting your revised manuscript for review.

I agree that some of my previous comments have been addressed; however, some remain outstanding:

1. The synthesis method has not been clarified. By synthesis, I mean how the quantitative and qualitative data interact/complement or oppose.
2. The reproductive age has not been added in the abstract (line 11).
3. In terms of my query on double reporting by including systematic reviews and meta-analysis in the dataset, this is still unclear. Although you have added lines 125-127, there is still a sentence in line 83 on page five that suggests that these are included and this is supported by another line in the exclusion criteria section.
4. Your justification for the search date limitation to make your dataset more 'manageable' and 'current' is inappropriate. What if you have missed seminal research for example?

Also:

5. Lines 14-15 say 'quantitative' twice. I think that the second should say 'qualitative'.
6. Please add a reference for the PCC framework.

Reviewer #2

(Remarks to the Author)

I have no further comments, following revisions made by the authors.

Version 2:

Reviewer comments:

Reviewer #1

(Remarks to the Author)

This is my review of a second revision, and the authors have comprehensively answered my queries.

I would like to thank the authors for their response, and I have nothing further to add.

Addressing reviewers' comments on Contraceptive-induced menstrual changes in low- and middle-income countries: a systematic scoping review

Reviewer #1	
This is a scoping review of the existing literature and as such, none of the claims are 'novel'; however, the review does bring together the evidence to allow an understanding/overview of the current status of the effect of CMIC on contraceptive use in LMICs.	Thank you for your feedback.
It is disappointing that the qualitative data is not presented, apart from a list of themes. As a mixed methods review, the reporting should pay as much attention to the qualitative findings as the quantitative findings and there should also be a further synthesis of how the two paradigms resonate together.	Thank you for your valuable comment. As suggested, we have expanded the qualitative synthesis. We have also highlighted how the quantitative and qualitative findings resonate together. However, it is worth noting that this is not a mixed-methods systematic review but a scoping review that explored four key concepts. Of the four concepts, only one (health impact of contraceptive-induced menstrual changes) included qualitative findings. Lines 174 – 219; Pages 8 - 10
I am unsure that the paper will influence thinking in the field, although it does provide an overview of the current situation on which to base future research.	Thank you for your comment. As a scoping review, this paper maps out the current gaps in research and will guide future research.
Abstract: Please clarify your synthesis method. In line 11 delete 'women' as this is duplicated.	The synthesis method has been clarified in the abstract and 'women' in line 11 deleted. "Quantitative data were summarised using descriptive statistics and quantitative data using thematic synthesis." Lines 14 – 15, page 2
Methods:	
1. Add the reproductive age range in line 11	The age range has been added "The participants/population were reproductive-age women (15 – 49 years) using any form of modern contraceptive (excluding barrier and permanent methods)"

	as defined by the source articles.” Lines 74 – 76, page 4
2. How did you ensure that there was no double reporting due to the inclusion of systematic reviews in your dataset? Surely some of these 16 reviews would have also included empirical studies you retrieved in your search?	We have clarified this in the methods section as follows: “To avoid duplication of data, systematic reviews were excluded from the data analysis; however, we reported the number of systematic reviews focused on each of the key concepts. Lines 125 – 127, page 6 In the results, we clarified this referring to the studies analysed as “primary studies”
3. Did you use a framework for your search strategy eg PICO?	This has been clarified: “The search strategy was formulated using the Participants/Population, Concept and Context (PCC) framework. The participants/population were reproductive-age women (15 – 49 years) using any form of modern contraceptive (excluding barrier and permanent methods) as defined by the source articles. The concepts explored included the prevalence, health impact, treatment and barriers to accessing treatment. The context considered included studies in countries classified as low-income, lower-middle-income or upper-middle-income according to the 2022 World Bank classification of countries by income level.” Lines 73 – 79, page 4 - 5
4. What is the justification for your search start date of 1st January 2000?	We have added a justification for the search date as follows: “The search was limited to this timeframe to ensure a manageable review that reflects current evidence.” Lines 98 – 99, Page 5
Findings:	
1. 321 studies is very high and perhaps resulted in the lack of presentation of qualitative findings?	The number of studies included is high because this is a scoping review and is not a mixed-methods systematic review. The review aimed to systematically map the existing literature to identify the scope of evidence on contraceptive-induced menstrual bleeding and gaps in research. We have expanded on the qualitative findings.

2. The qualitative synthesis needs attention - a list of themes is not appropriate in a published review.	We have expanded on the qualitative synthesis by providing a more in-depth description of the themes identified. Lines 174 – 219; Pages 8 - 10
Discussion	
1. The 2009 review discussed in lines 205-211 - is this one of the included reviews in your scoping review or is it meant to be supporting literature? It isn't clear and doesn't seem to resonate with the previous point.	Thank you for your valuable feedback. The paragraph has been rephrased and the reference updated as follows: “...Sub-analyses of studies using standard definitions and measurements of CIMC with at least 12 months of follow-up in LMICs yielded less variable prevalence estimates. This highlights the importance of using standardised definitions and measurements of CIMC in primary studies to improve comparison between studies and optimise CIMC data.²⁹ Optimising data will support counselling and guide decision making regarding contraceptive choice.²⁹” Lines 243 – 248, page 11
2. Please provide the limitations of your review as well as the strengths	This is included on page 14 (lines 305 – 317)
3. Please provide a summary of the implications of your review for clinical practice	We have added a subheading to highlight this in the discussion section. Lines 263 – 292; Pages 12 - 13
4. Please provide a summary of recommendations for future research	We have added a subheading to highlight this in the discussion section. Lines 293 – 304; pages 13 - 14
Tables:	
There is no table of included study characteristics which is good practice in a review	We presented study characteristics in figures and tables in the supplementary material. Considering that this is a scoping review exploring four key concepts (prevalence, burden, treatment and barriers to accessing treatment for contraceptive-induced menstrual changes), it did not seem reasonable to have a table with 321 individual studies listed in the main paper. We have added a spreadsheet containing the main characteristics of the included studies (Supplementary file 4)
I do not feel that there is sufficient methods detail to reproduce this study.	We have added a more detailed search framework. The search strategy for each database is also included in the

	supplementary materials to ensure reproducibility.
Reviewer #4	
The paper reports on findings from a systematic scoping review on menstrual changes associated with contraception. The review draws from a large data set of 321 studies from 44 low- and medium-income countries (LMIC) on contraceptive dissatisfaction and subsequent method discontinuation, and how this contributes to a high unmet need for contraception in LMIC. The prevalence of contraceptive-induced menstrual changes varied from 0-94%, depending by contraceptive type. In this context, the review paper calls for greater investment in research and development of therapeutics and novel approaches to reduce contraceptive-induced menstrual changes. Changes documented include prolonged or frequent or irregular bleeding, The findings from the review make a compelling case for addressing contraceptive-induced menstrual changes.	Thank you for your valuable feedback.
From their deduction from the findings, the authors allude to method discontinuation due to contraceptive-induced menstrual changes as the most frequently reported "the burden of contraceptive-induced menstrual changes". Should this rather be "outcome associated with contraceptive-induced menstrual changes?"	Thank you for your feedback. We agree that the term 'burden' does not accurately portray our meaning and have changed this throughout the manuscript to 'health impact'.

Addressing reviewer’s comments on Contraceptive-induced menstrual changes in low- and middle-income countries: a systematic scoping review

Reviewer #1	
1) The synthesis method has not been clarified. By synthesis, I mean how the quantitative and qualitative data interact/complement or oppose.	Thank you for your feedback. We have clarified this as follows: “We employed an explanatory sequential synthesis method to integrate quantitative and qualitative findings. First, quantitative findings were synthesised using descriptive statistics (frequency and percentages) in Microsoft Excel to summarise key trends and patterns. Next, following previously published methods, two reviewers (MM and JC) conducted a thematic synthesis of qualitative findings. This involved line-by-line coding of the extracted data, which were organised into initial codes and iteratively refined to inductively generate and define overarching themes.³² The qualitative findings were then used to elaborate on and contextualise the quantitative findings. Integration was conducted through a narrative approach, highlighting areas where qualitative insights clarified or explained trends and patterns observed in the quantitative data.” Pages 8 Lines 142 - 151
2) The reproductive age has not been added in the abstract (line 11)	This has now been added to the abstract as well. (15 – 49 years) Page 2 Line 11
3) In terms of my query on double reporting by including systematic reviews and meta-analysis in the dataset, this is still unclear. Although you have added lines 125-127, there is still a sentence in line 83 on page five that suggests that these are included, and this is supported by another line in the exclusion criteria section.	Thank you for your comment. The sentence in the eligibility criteria is: ‘We included primary research using the following study designs: interventional trials (randomised and non-randomised trials); observational studies (prospective cohort, retrospective, cross-sectional, and case-control studies); qualitative studies; mixed methods studies; and review of primary research (only systematic reviews and meta-analysis)’.

	This is correct, we included systematic reviews in our scoping review, but we did not include it in the main analysis as we did the primary studies. The data extracted from systematic reviews were only related to the key concepts captured by the reviews and were not combined with the analysis of the primary data. Systematic reviews were included to ensure that we captured the breadth of evidence relevant to the key concepts of our scoping review. We have rephrased this for clarity as follows: 'To avoid duplication of data, systematic reviews were excluded from the main analysis. However, we reported the number of systematic reviews addressing each of the key concepts to provide an overview of the evidence base.' Page 8, Lines 151 - 153
4) Your justification for the search date limitation to make your dataset more 'manageable' and 'current' is inappropriate. What if you have missed seminal research for example?	Thank you for your feedback, we have revised this with consideration for your concern: 'We limited the search to studies published from 2000 onwards to ensure the review captured the most recent and relevant evidence. This timeframe reflects a period of significant innovation in contraceptive technologies, including the wider adoption of long-acting reversible contraceptives. While this approach may exclude some earlier seminal studies, our focus was on synthesising evidence aligned with current clinical practice and policy priorities.' Page 7 Lines 115 - 120
5) Lines 14-15 say 'quantitative' twice. I think that the second should say 'qualitative'.	Thank you for drawing attention to this, we have corrected it.
6) Please add a reference for the PCC framework.	We have added the reference